# Antioxidant Properties of Hemp Proteins: From Functional Food to Phytotherapy and Beyond

**DOI:** 10.3390/molecules27227924

**Published:** 2022-11-16

**Authors:** Jiejia Zhang, Jason Griffin, Yonghui Li, Donghai Wang, Weiqun Wang

**Affiliations:** 1Department of Food, Nutrition, Dietetics and Health, Kansas State University, Manhattan, KS 66506, USA; 2John C. Pair Horticultural Center, Department of Horticulture & Natural Resources, Kansas State University, Haysville, KS 67060, USA; 3Department of Grain Science and Industry, Kansas State University, Manhattan, KS 66506, USA; 4Department of Biological and Agricultural Engineering, Kansas State University, Manhattan, KS 66506, USA

**Keywords:** hemp, hemp peptide, antioxidant activity, structural characteristic, pathogenesis-related molecular mechanism

## Abstract

As one of the oldest plants cultivated by humans, hemp used to be banned in the United States but returned as a legal crop in 2018. Since then, the United States has become the leading hemp producer in the world. Currently, hemp attracts increasing attention from consumers and scientists as hemp products provide a wide spectrum of potential functions. Particularly, bioactive peptides derived from hemp proteins have been proven to be strong antioxidants, which is an extremely hot research topic in recent years. However, some controversial disputes and unknown issues are still underway to be explored and verified in the aspects of technique, methodology, characteristic, mechanism, application, caution, etc. Therefore, this review focusing on the antioxidant properties of hemp proteins is necessary to discuss the multiple critical issues, including in vitro structure-modifying techniques and antioxidant assays, structure-activity relationships of antioxidant peptides, pre-clinical studies on hemp proteins and pathogenesis-related molecular mechanisms, usage and potential hazard, and novel advanced techniques involving bioinformatics methodology (QSAR, PPI, GO, KEGG), proteomic analysis, and genomics analysis, etc. Taken together, the antioxidant potential of hemp proteins may provide both functional food benefits and phytotherapy efficacy to human health.

## 1. Introduction

Excessive oxidation is one of the primary causes of mitochondrial damage, DNA damage, cell pyroptosis, carcinogenesis, and chronic diseases [1,2,3,4]. Thus, the biological system needs antioxidant defenses to protect the body against oxidative attacks [3]. In general, antioxidants can be classified into three categories: (1) Hydrophilic and lipophilic antioxidants [5]; (2) Endogenous and exogenous antioxidants [6,7,8]; and (3) Synthetic and natural antioxidants. In recent decades, some antioxidants are being developed as therapeutic agents, nutrition supplements, and food additives to enhance human health [9,10,11,12,13,14]. However, the safety of antioxidants has been questioned. In a typical example, butylated hydroxytoluene at high doses could induce the non-apoptotic cell death of thymocytes in rat models [15,16]. Therefore, natural exogenous antioxidants such as dietary hemp protein have attracted more attention over the past few years.

Throughout history, hemp is one of the oldest plants cultivated by humans [17]. It was found around 800 B.C. Hemp used to be a legal crop before the nineteenth century in the USA. However, the prohibition of hemp cultivation was imposed on the hemp industry in 1970, because hemp contains two neurotoxic ingredients, cannabidiol and delta-9-tetrahydro-cannabinol, which were unable to be legally described [18]. Hemp returned to the USA. as a completely legal crop till the Agricultural Improvement of 2018 was signed by the USA president. As a result, it has stimulated an increasing number of researchers to conduct various novel studies on hemp [19,20,21].

While bioactive peptides derived from hemp proteins have been proven to be strong antioxidants, some controversial disputes and unknown issues are still underway to be explored and verified in the aspects of technique, methodology, characteristic, mechanism, application, caution, etc. Therefore, this review focusing on the antioxidant properties of hemp proteins aims to discuss the multiple critical issues, including in vitro structure-modifying techniques and antioxidant assays, structure-activity relationships of antioxidant peptides, pre-clinical studies on hemp proteins and pathogenesis-related molecular mechanisms, usage and potential hazard, and novel advanced techniques. The antioxidant potential of hemp proteins may provide both functional food benefits and phytotherapy efficacy to human health.

## 2. Structure-Modifying Techniques

To better meet bioactive needs, complex protein structures need to be converted into simple subunits, and short-chained peptides have to be separated from the aggregate mass [22,23]. Usually, two effective and economical approaches, pH-shift and enzymatic hydrolysis are employed to release specific peptides from protein.

### 2.1. pH-Shift

pH-shift is a straightforward structure-modifying method. When a protein is incubated with an extreme alkaline or/and acidic, the structure of the protein is unfolded or/and refolded so that more functional structures can be exposed [22,23].

### 2.2. Enzymatic Hydrolysis

In general, bioactive peptides are idle when they are encoded in the parent proteins. However, once proteins are hydrolyzed by enzymes, they are able to play more physiological roles in human health [24,25,26]. Clearly, the efficiency of hydrolysis can be impacted by multiple factors such as the type of enzymes, hydrolysis time, pH value, temperature, enzyme-substrate ratio, and sample-water ratio. Among these factors, enzyme specificity is the most important factor in the cleavage of peptide chains [27,28]. Then, the degree of hydrolysis (DH) of the cleaved peptides can usually be evaluated by the O-phthaldialdehyde (OPA) assay [28,29]. In the nutritional study conducted by Aiello et al. (2017), hemp protein was hydrolyzed by pancreatin, pepsin, trypsin, and a mixture of enzymes (co-digested), respectively. The highest DH of hemp protein was achieved by pancreatin (47.5%), followed by trypsin (46.6%), co-digested (34%), and pepsin (19.7%) [30]. In another study, the antioxidant activities of hemp protein hydrolyzed by six different digestive enzymes (pepsin, alcalase, neutrase, flavourzyme, trypsin, and protamex) were tested. The results implied that enzyme specificity is one of the determinants for high-value antioxidant products of hemp protein [31].

## 3. Assessments of the Antioxidant Ability

A series of mature techniques such as reactive oxygen species (ROS) scavenging ability assay, metal ion chelating capacity assay, and assessments of antioxidant enzymatic and non-enzymatic activities have been constantly developed and widely applied to evaluate the antioxidant activity of various natural proteins. Through these techniques, a large amount of research work has made considerable contributions to the identification of bioactive antioxidant peptides [32].

### 3.1. Reactive Oxygen Species Scavenging Ability

ROS including superoxide, hydroxyl, and peroxy radicals are characterized as a cluster of oxidants or free radicals [3]. Free radical scavenging ability is one of the essential functions of antioxidants, which is associated with hydrogen atom transfer (HAT) and electron transfer (ET) [3,32].

With the regard to the antioxidant ability assays caused by HAT-based reactions, the Oxygen Radical Absorbance Capacity (ORAC) method is a common method to assess antioxidant activities [3,33,34]. The mechanism of the ORAC assay is as follows. One 2,2′-azobis (2-amidinopropane) dihydrochloride (AAPH) molecule loses the dinitrogen (R-N=N-R) to produce two radicals (R**^·^**). Under an O_2_-saturated condition, the radical abstracts an O_2_ to generate a peroxyl radical (ROO**^·^**). ROO**^·^** reacts with a hydrogen atom (XOH) to form hydroperoxide (ROOH) and an antioxidant radical (XO^·^) (R-N=N-R→2R**^·^** + N_2_; R^·^+ O_2_ →ROO**^·^**; ROO**^·^** + XOH→ROOH + XO**^·^**) [35,36,37,38].

Similarly, other standard antioxidant ability assays are usually employed to analyze the antioxidant ability triggered by ET-based reactions, such as 2,2-diphenyl-1-picrylhydrazyl (DPPH) radical scavenging activity, Ferric Reduction Antioxidant Power (FRAP) assay, and Trolox Equivalent Antioxidant Capacity (TEAC) [3,33,34].

In a study of hemp antioxidant ability [39], Lu et al. (2010) reported that hemp protein hydrolyzed by alcalase for 1 h, had good DPPH radical scavenging ability (3.75 ± 0.09 mg/mL), superoxide radical scavenging ability (3.5 ± 0.1 mg/mL), hydroxyl radical scavenging ability (7.25 ± 0.3 mg/mL), and FRAP ability (0.21 ± 0.03 mg/mL) [39]. Likewise, the results of Frassinetti et al. displayed that hemp seed extracts had high ORAC activity (127 ± 5 µmol TE/g), and DPPH radical scavenging activity (40 ± 2%) [40]. The likely explanations for these results were that hemp antioxidant peptides contained the donors of electrons and hydrogens, which could interrupt the radical-chain reactions, so they could convert free radicals to stable species [39,40].

The free radical scavenging ability assays described above had already been extended in the hemp application. For instance, a novel gluten-free cracker fortified with hemp seed oil pressed-cake and decaffeinated green tea leaves was developed. The antioxidant property of crackers was tested by DPPH (30.3 μmol TE/g d.w.). The overall score of 8.9 indicated it had good antioxidant properties. Moreover, the researchers further interpreted that the crackers still had antioxidant properties even without green tea. They implied that hemp seed as a nutritional food additive had promising applications in snack marking [41].

### 3.2. Metal Ion Chelating Capacity

Additionally, metal (Fe) ion-chelating is a classical assay to assess the antioxidant ability of protein. Its chemical mechanism is related to Fenton reactions (Fe^2+^ + H_2_O_2_→Fe^3+^ + OH^−^ + OH; 2Fe^3+^ + H_2_O_2_→2Fe^2+^ + O_2_ + 2H^+^; Fe^2+^ + O_2_→Fe^3+^ + O_2_^−^). Since iron has high reactivity, ferrous chelators can minimize Fe^2+^, thereby disrupting harmful metal-catalyzed oxidation and reducing the production of oxygen-free radicals [42,43]. Girgih et al. (2011) tested the antioxidant ability of hemp protein hydrolysate (alcalase) and found that hemp hydrolysate had a strong metal ion chelating ability (72%) [44]. Similarly, the work of Wang et al. elucidated that hemp protein hydrolysate (neutrase) had a high metal ion chelating ability (1.7–1.8 mg/mL) and DPPH radical scavenging activity (2.3–2.4 mg/mL) [45].

### 3.3. Assessments of Antioxidant Enzymatic and Non-Enzymatic Activities

The human body has its own endogenous antioxidants to defend against the damage of ROS. These antioxidant-related mechanisms possess two arms, enzymatic antioxidants and non-enzymatic antioxidants.

Enzymatic antioxidants including glutathione peroxidase (GPx), catalase (CAT) and superoxide dismutase (SOD) can prevent ROS damage by converting superoxide radicals to stable compounds. Due to this mechanism, they are favorable biological indicators of antioxidant activity. Take SOD as a good example. The superoxide is formed from an oxygen atom and an electron (O_2_ + electron→O_2_^·^). It is a long-lived molecule that can diffuse readily. The transition from superoxide to hydrogenous oxide is catalyzed by SOD. Therefore, SOD activity can reflect the state of superoxide under the actions of antioxidants [46,47]. In a cell culture experiment, Hong et al. (2015) illustrated that hemp seed extracted by ethanol and supercritical fluid could enhance the gene expression of antioxidant enzymes (SOD, CAT, and GPx) in HepG2 cells [48]. Another in vivo experiment conducted by Girgih, Abraham T. et al. demonstrated that the plasma SOD and CAT activities in spontaneously hypertensive rats (SHRs) consuming the hemp diet were significantly higher than those in the control group [46,49].

Non-enzymatic antioxidants including glutathione (GSH), bilirubin, α-tocopherol, and ascorbic acid can also provide defense against ROS, clear up free radicals, and maintain the balance of redox [46,47]. A relatively recent study by Kubiliene et al. (2021) indicated that hemp extract could significantly reduce the level of GSH (26.81%) in the brain and liver of mice [50]. Similarly, a decreasing concentration of GSH was detected in the splenocyte cells treated with hemp derivatives [51]. These results supported that hemp could work well as an antioxidant.

So far, we also completed a pilot experiment on the antioxidant ability of hemp protein. In our pilot experiment, the hemp protein and soybean (control) were hydrolyzed by neutrase and the hydrolyzates were further assessed by the ORAC method modified from Ou et al. [52] and the metal iron chelating activity assay modified from Elias et al. [53]. The results in Table 1 showed that the antioxidant activities of hemp protein hydrolysate were significantly higher than that of soybean protein hydrolysate (Table 1). This pilot experiment paves the way for future in vitro and in vivo studies involving cell and small animal models to explore more biological significance of hemp protein.

## 4. Structure-Activity Relationship of Antioxidant Peptides

Considering that the antioxidant activities of bioactive peptides are mainly associated with the molecular weight (MW) of peptides, the compositions of amino acids, and the sequences of amino acid groups [54,55,56], the structure-activity relationship of antioxidant peptides can be studied by the two techniques below.

The first technique includes three traditional methods, membrane ultrafiltration (MU), mass spectrometry (MS), and reverse-phase high-performance liquid chromatography (RP-HPLC). They are extensively used to detect the relationship between the peptide structure and antioxidant activity through the isolation and identification of peptides. Membrane ultrafiltration separates the small MW of peptides from the large MW of peptides by membranes with specific pore sizes [54]. MS analyzes the compositions of amino acids using the optics principles [57,58]. RP-HPLC isolates bioactive peptides based on the hydrophobic characteristics of peptide [30,59]. In practice, the hemp hydrolysates can be filtered by MU first. Subsequently, the low MW of hemp hydrolysate can be tested through in vitro and in vivo assays to detect its antioxidant abilities. Finally, the compositions and structures of antioxidant peptides are verified by RP-HPLC and MS. This experimental flow chart of antioxidant peptides is shown in Figure 1 [32].

The second technique is a novel bioinformatics method, quantitative structure-activity relationship (QSAR) statistical model, which can predict the biological efficacy of peptides. Although there are a lot of existing studies on the QSAR of anti-hypertensive peptides, the number of studies on the QSAR of antioxidant peptides is far less than that of anti-hypertensive peptides. This may be because antioxidant peptides have many dependent, independent, and control variables, making the statistical analysis of antioxidant peptides extremely complicated [60,61,62,63,64,65].

### 4.1. Molecular Weight

Generally, the short-chain peptides derived from food protein have several physiological benefits such as promoting the gastrointestinal digestion process, being more absorbed into the blood circulation to reach target organs rapidly and preventing oxidative damage [66,67,68,69]. Hence, Girgih et al. (2011) suspected that peptides with appropriately low MW had strong antioxidant abilities. In their study, the hemp seed protein was hydrolyzed by alcalase and separated by MU to obtain the low MW of peptide fractions, e.g., >5, 3–5, 1–3, and <1 kDa MW of peptides. The <1 kDa fractions demonstrated the highest DPPH radical scavenging ability (24.2%) [44].

### 4.2. Composition of Amino Acids

Notably, the composition of amino acids is a determining factor in the antioxidant activities of peptides. Amino acids can be divided into hydrophobic amino acids (Ala, Leu, Met, Ile, Pro, Phe, Trp, Val) and hydrophilic amino acids. The hydrophilic amino acids can be subclassified into neutral amino acids (Gly, Thr, Tyr Ser, Cys, Gln, Asn), acidic amino acids (Asp, Glu), and alkaline amino acids (Lys, Arg, His) [70,71,72].

The hydrophobic amino acids serve as great hydroxyl radical scavengers owing to their indole structure and the benzene ring, which can donate hydrogen in lipid peroxidation reactions (RH + OH**^·^** →R**^·^** + H_2_O; R**^·^** + O_2_→ROO**^·^**; ROO**^·^** + RH→ROOH + R**^·^**) [3,27,54]. They deter lipid peroxidation reactions by converting hydrogen from the indole structure and benzene ring to free radicals. That may be the reason why the hydrophobic amino acids have a strong antioxidant ability [70,71,72].

For example, the peptides from tuna muscle exhibited strong antioxidant ability due to their high proportion of hydrophobic amino acids [73]. Likewise, Chi et al. (2015) analyzed bluefin protein and verified that peptides (Trp-Glu-Gly-Pro-Lys, Gly-Pro-Pro, Gly-Val-Pro-Leu-Thr) containing high hydrophobic amino acid content could work well as natural antioxidants [74].

On the contrary, other scholars do not agree with above-mentioned perspective. They think that hydrophilic amino acids, especially neutral amino and alkaline amino acids may have good performance in antioxidant activities. For instance, Chen et al. (1996) discovered that the bioactive peptide chains containing His were credited with good metal ion chelating ability due to the presence of α-amido, carboxyl, and imidazole structures [66]. Moreover, Tian et al. (2005) predicted Cys was one of the most active antioxidant amino acids by a QSAR model. It might be associated with the number of strong disulfide bonds in a precursor molecule [75].

Additionally, the amino acid contents of hemp protein can be modified by physical and chemical techniques. Table 2 exhibits that the amino acid contents of defatted hemp seed, hemp protein hydrolysate, hemp protein isolations, and hemp protein fractions are different, which verifies this viewpoint. Although the underlying mechanism is less clear, the results of hemp protein studies listed in Table 2 can provide scholars with useful information for reference. For example, the Arg content of hemp protein was the highest by enzymatic hydrolysis treatment (pepsin and pancreatin) and the lowest by pH-shift treatment (acid extraction). If the Arg content of hemp is studied, acid extraction might be avoided.

### 4.3. Sequence of Amino Acids

Importantly, some scientists found that the amino acids of the same composition did not have similar bioactive effects. They deemed that the sequence of the amino acids played a significant role in antioxidant ability [55,56]. Despite the controversy, the following studies are still good examples to interpret how the sequence of amino acids affects the antioxidant activities of specific peptides. Chen et al. (1996) discovered that the metal ion chelating capacity of His-Leu was higher than that of Leu-His, the same as His-His-Pro > Pro-His-His, His-His-Leu-Pro > Pro-Leu-His-His, and His-His-Pro-Leu-Leu > Leu-Leu-Pro-His-His. Apparently, His at the N-terminus of antioxidant peptides had better metal ion chelating capacity [66]. In contrast, other scholars consider that the peptides containing hydrophobic amino acids at the N-terminal and hydrophilic amino acids at the C-terminal have high free radical scavenging ability. Bougatef and his coworkers studied seven peptides from ardinella, Pro-His-Tyr-Leu, Gly-Ala-His, Gly-Ala-Leu-Ala-Ala-His, Leu-His-Tyr, Gly-Gly-Glu, Leu-Ala-Arg-Leu, and Gly-Ala-Trp-Ala. They stated that the Leu-His-Tyr had the highest antioxidant ability (DPPH 63 ± 1.57%) [78].

Specially, Xu et al. (2004) studied the antioxidant peptide, Leu-Asp-Tyr-Glu from corn. In their opinion, the reason why this peptide had a strong antioxidant ability was that Tyr was adjacent to Asp and Glu, so the hydrogen donating capacity of Tyr could be enhanced. The carboxylate of Asp and Glu could attract electrons, and then the density of the electron cloud in phenolic hydroxyl of Tyr was decreased. These reactions made it easier for hydrogen of Tyr to be released [79]. Similar results had been observed by Bamdad et al. (2005) that among the six peptides from barley protein, Pro-Tyr-Pro, Gln-Pro-Tyr-Pro-Gln, Gln-Gln-Pro-Tyr-Pro-Gln, Gln-Pro-Gln-Pro-Tyr-Pro-Gln, Thr-Gln-Gln-Pro-Tyr-Pro-Gln, and Glu-Pro-Tyr-Pro-Glu, Gln-Pro-Tyr-Pro-Gln had the strongest antioxidant ability since it contained symmetrically repeated sequences. The Gln and Pro near Tyr could help release the hydrogen in the phenolic hydroxyl of Tyr, boosting the antioxidant ability of peptides [80].

Surprisingly, in alignment with these results, a recent QSAR study also observed that Tyr, Trp, or Cys located at the C-terminal position played a more dominant role in antioxidant activity than other C-terminal residues. Based on the predicted antioxidant values of 7870 unknown tripeptides, Du et al. compared 14 popular machine learning methods for the QSAR models and then screened tripeptides to identify critical amino acid features that determine the antioxidant activity. Later, they found that potentially high antioxidant activity tripeptides all have a Tyr, Trp, or Cys residue at the C-terminal position [65].

Supplementarily, Table 3 shows the amino acid sequence of nutritious food with antioxidant ability, which can also support the above viewpoints. However, the structure-activity relationship of antioxidant peptides has not been completely understood. More studies on the advanced structure-modifying techniques such as glycosylation and high-intensity ultrasound, the structure identification of antioxidant peptides, as well as the correlation between antioxidant protein and other antioxidant phytochemicals that can fortify protein function such as polyphenols are required to expand the utilization of the hemp protein as a functional food [81,82,83,84,85].

## 5. Pre-Clinical Studies on Hemp Seed and Pathogenesis-Related Molecular Mechanisms

Moreover, health professionals have been increasingly interested in the phytotherapy efficacy of hemp seed such as anti-inflammatory, anticancer, and anti-hypertension due to its strong antioxidant ability. This offers a wide range of potential pharmacology for chronic diseases involving inflammatory diseases, cancer, and hypertension [7,40,67,68,91,92,93,94,95].

### 5.1. Anti-Hypertension Effect

The rennin-angiotensin-aldosterone system (RAAS) is the primary vasopressor system in human beings. The angiotensinogen produced from the liver is firstly cleaved by renal renin to generate angiotensin I. Next, the angiotensin-converting enzyme (ACE) removes the C-terminal dipeptide of angiotensin I to form angiotensin II, an effective vasoconstrictor peptide. Subsequently, increasing angiotensin II levels enables the nitric oxide synthase (NOS) to convert L-arginine into L-arginine and nitric oxide (NO), which is a potent vasodilator [49,96,97]. Furthermore, excessive ROS affects the levels of Asymmetric dimethylarginine (ADMA), an endogenous inhibitor of NOS. When ADMA reaches a certain level, peroxynitrite can be produced by uncoupled NOS isoenzymes, causing a decrease in NO bioavailability. These reactions lead to NO disequilibrium and then induce high blood pressure. Therefore, maintaining the balance of ROS and NO is an effective strategy for hypertension treatment (Figure 2) [98].

In these metabolic pathways, hemp protein hydrolysate (HPH) can play an anti-hypertensive role due to its peptide characteristics such as the high contents of arginine and the bioactive peptides (e.g., Trp-Tyr-Thr, Ser-Val-Tyr-Thr, Ile-Pro-Ala-Gly-Val), which can increase the No levels, reduce oxidative stress, and suppress the renin and ACE activities [49,97,99] carried out two animal experiments and suggested that hemp products had both anti-hypertension and antioxidant effects. Firstly, 36 spontaneously hypertensive rats (SHRs) were treated with saline, captopril, and hemp protein, respectively. The systolic blood pressure (SBP), plasma ACE activity, and renin level of SHRs treated with hemp hydrolysate were lower than those in the control group. Subsequently, another rat experiment was conducted to identify the antioxidant ability of hemp protein. 32 SHRs were equally divided into three groups. They were fed with diets containing 1.0%, 0.5% and 0% (*w*/*w*) hemp hydrolysate, respectively. The plasma SOD and CAT activities in SHRs fed the hemp diets were significantly higher than those in the control group [49,97]. Thus, they suggested that the antioxidant peptides derived from hemp seed might be a promising therapeutic agent for hypertension since these hemp bioactive peptides could improve the activities of antioxidant enzymes, lower ROS and oxidative stress and finally reduce the level of ACE, renin, and blood pressure [44,49,97,100].

### 5.2. Anti-Inflammatory Effect

In theory, NF-κB (Nuclear Factor κB) dimer is a pleiotropic transcription factor, so the NF-κB signaling pathway can regulate the expression of many genes on inflammation. NF-κB exists in the cells in the form of the NF-κB-IκB (Inhibitor of NF-κB) complex. When ROSs attack the cells, IKKβ (inhibitor of κB Kinase β) can be phosphorylated and activated. Next, NF-κB dimer can be released from NF-κB-IκB by phosphorylated IKKβ. NF-κB dimer contains two critical DNA binding domains, the N-terminal Rel homology domain (RHD) and the C-terminal transcription activation domain (TAD). Once NF-κB no longer binds with IκB, RHD and TAD can be exposed. Free NF-κB dimer is further activated by multiple posttranslational factors. Meanwhile, NF-κB dimer relocates to the nucleus and binds to specific DNA sequences that can match with RHD and TAD such as some promoters and enhancers. These reactions boost transcriptions of proinflammatory genes and generate inflammatory cytokines such as tumor necrosis factor (TNF-α), interleukin (IL)-1β, and IL-6 (Figure 3) [76,101,102,103,104,105,106,107].

Keap1-Nrf2 (Keap1, Kelch-like ECH-associated protein 1; Nrf2, nuclear factor erythroid 2-related factor 2) is another vital antioxidant and anti-inflammatory signaling pathway. In this signaling pathway, hemp bioactive peptides can promote the uncoupling of the Keap1-Nrf2 complex [108,109,110]. On the one hand, the released Nrf2s enter the cell nucleus and bind to antioxidant response elements (ARE), upregulating the gene expression of various antioxidant enzymes including Heme Oxygenase-1 (HO-1), Nicotinamide Adenine Dinucleotide Phosphate Quinone Oxidoreductase 1 (NQO), SOD, GSH-Px, and CAT, to improve the antioxidant activities. On the other hand, the released Keap1 can bind to IKKβ to inhibit the activities of IKKβ so that the NF-κB-IκB complex would be steady. Consequently, the progression of inflammation may be slowed down due to the low levels of inflammatory cytokines (Figure 3) [111,112,113,114,115,116].

Using BV-2 microglial cell model, Noelia et al. reported that hemp has potential pharmaceutical value on anti-inflammatory effect. The defatted hemp meal was initially hydrolyzed by flavourzyme and alcalase under multiple conditions. Among the bioactive hemp protein hydrolysates, HPH20A (alcalase for 20 min) and HPH60A + 15AF (alcalase for 60 min and flavourzyme for 15 min) were selected for further experiment of cell model because of their high DH values. The lipopolysaccharide (LPS)-stimulated BV-2 microglial cells were treated with HPH60A + 15AF and HPH20A at 50 and 100 μg/mL-1, respectively. The results indicated that both HPH60A + 15AF and HPH20A could up-regulate the gene expression of interleukin (IL)-10 and suppress the mRNA transcriptional levels of IL-1β, TNF-α, and IL-6 in BV2 microglia cell [76]. Moreover, in (LPS)-stimulated BV2 microglia cells, Wang et al. (2019) also found that hemp seed (cannabisin) could suppress the mRNA levels of TNF-α and IL-6. More importantly, the expression level of Sirtuin 1(SIRT1), which can attenuate inflammation was improved since the inhibitor of SIRT1 was suppressed. At the same time, as Nrf2 was released under the effect of hemp derivant, the expression level of HO-1 was increased, and then ROS was reduced. Thus, hemp seed had both anti-inflammatory and antioxidant abilities to inhibit inflammatory progression related to NF-κB and Keap1-Nrf2 signaling pathways [117].

### 5.3. Anti-Cancer Effect

Strictly speaking, Akt (Protein kinase B)/GSK-3β (Glycogen synthase kinase-3β)/β-catenin signaling pathway is recently discovered for the etiology of cancer diseases. β-catenin, a transcription factor, can improve the transcription levels of oncology genes and promote the progression of cancer. Wei et al. proposed a hypothesis that HPH might have an anti-cancer ability by suppressing Akt/GSK-3β/β-catenin signaling pathway. HPH is likely to activate Akt, which then phosphorylates and inactivates GSK-3β. Subsequently, β-catenin can be phosphorylated by inactive GSK-3β. The phosphorylated β-catenin is further ubiquitinated and degraded. As a result, less β-catenin can be released from the complex and enter the cell nucleus. Lastly, the gene expressions of oncology genes are inhibited (Figure 4) [118,119,120,121,122,123,124].

In the study of Wei et al. (2021), hemp proteins hydrolyzed by neutral protease and papain was filtered by Ultrafiltration first. Subsequently, hepatocyte tumor cells (Hep3B) and normal hepatocyte cells (L02) were treated with the low MW of HPH for 24 h. In Hep3B cells, the cellular viability, the cell proliferation, the cell migration, the levels of Bcl-2 (anti-apoptotic protein), and the phosphorylation of Akt were decreased. Meanwhile, Bax (pro-apoptotic protein), caspase 3 (pro-apoptotic protein), as well as the phosphorylation of GSK-3β and β-catenin were increased. However, in L02 cells, there were no statistical differences. These results verified that HPH could promote the apoptosis progress of cancer cell through Akt/GSK3β/β-catenin signaling pathway [123].

Another good experiment on the anti-cancer ability of hemp seed protein was conducted by Lu et al. (2010). Hemp seed protein was hydrolyzed by alcalase to generate bioactive ingredients. These bioactive ingredients were separated, purified, and identified by macroporous adsorption resin, gel filtration, RP-HPLC and MS to further obtain two antioxidant peptides, His-Val-Arg-Glu-Thr-Ala-Leu-Val and Asn-His-Ala-Val. The apoptosis levels of the hydrogen-peroxide-stimulated PC 12 cells treated with hemp peptides were evaluated. The results demonstrated the hydrogen peroxide-induced cell toxicities were significantly mitigated by hemp antioxidant peptides [39].

## 6. Hemp Seed Protein Product and Usage

Compared with other plant foods, hemp seed is a superior nutraceutical resource, since hemp seed protein has a balanced amino acid profile and contains a great amount of essential amino acids such as His, Arg, Met and Leu. Hemp seed can provide sufficient amino acids to meet the daily requirements of children according to the recommendations of the Food and Agriculture Organization (FAO), World Health Organization (WHO), and United Nations University (UNU) (Table 4) [125,126,127]. Therefore, hemp protein products can be recommended for people with low-protein intakes, vegans and vegetarians, lactose intolerant people, patients with osteoporosis, athletes, dieters, children, etc. In recent years, hemp protein has been directly added to daily diets through advanced food science technologies, such as bakery products, milk, and meats for health benefits [128,129,130,131]. For instance, hemp flour could be blended into gluten-free bread. It limited the recrystallization of amylopectin and the hardening of crumb (the main symptom of bread staling) during bread storage [132]. Meanwhile, gluten-free bread fortified with hemp provided a higher intake of macroelements and microelements, especially proteins and iron. 10% of hemp derivatives could keep the balance between nutraceutical value and bread quality [130]. Additionally, hemp derivatives were applied to meat products. The quality of pork loaves added with hemp ingredients was evaluated by Zajac et al. (2019). The contents of fiber, magnesium, manganese, iron, copper, and polyunsaturated fatty acids in meat products were elevated, but the microbial growth was not affected. It can be generalized that hemp-treated meat products are healthy foods [131]. Specifically, hemp as a vegetable protein might be a good way to improve nutritional level of yogurt. In reviewing a recent literature, the authors studied that the fortification of plant protein-mixed yogurt. The effects of 5 vegetable proteins (hemp, pea, soy, wheat, and pumpkin) as additives were determined by rheological, physicochemical, and sensory properties of yogurt before pasteurization [129]. Hemp protein had the lowest syneresis value and the high viscosity. Furthermore, the yogurt fortified with hemp proteins also displayed an increase in acidity. These results revealed that the addition of hemp had good performance in dairy products [129,133,134].

## 7. Potential Hazard

However, hemp is a mild allergen for special populations because it can directly trigger side effects such as wheal, pruritus, and severe symptoms caused by type I hypersensitivity [141,142]. A case report showed that a 30-year-old man who worked in hemp harvesting suffered from contact urticaria. It demonstrated that hemp might be a resource of occupational allergen [141,143]. Additionally, hemp is an indirect allergen. When consumed with other specific plant allergens, hemp would magnify the adverse reactions, which are called “cannabis-fruit” or “cannabis-vegetable” syndromes [142,144]. To improve beneficial functions and alleviate side effects, the experiments on hemp seed proteome have emerged in recent years. Scientists identified the proteomic profile of hemp seed by gel electrophoresis, MS. Followed by bioinformatics methods such as protein–protein interactomics (PPI) network analysis, Gene ontology (GO) and Kyoto encyclopedia of genes and genomes (KEGG) pathways analysis, the correlation between the proteomic characteristics of hemp seed and its related physiological functions involving energy regulation, metabolism, stress response, cytoskeleton, and binding, protein synthesis was studied (Figure 5) [145,146]. Fortunately, two major hemp allergens, thaumatin-like protein and lipid transfer protein had been found and they could be purged by the food processing techniques [147]. The more negative properties of hemp seed protein can be discovered and eliminated, the more beneficial hemp seed protein will be.

Finally, protein is one of the most essential macronutrients for human beings. Since the excessive intake of animal protein is associated with a high risk of chronic diseases, hemp protein products gradually become a good alternative to animal protein. However, excessive protein may bring about side effects because the body dose not store proteins. Long-term and high-protein intake may increase the renal glomerular filtration rate and impose a significant metabolic burden on kidneys, thus leading to kidney damage, especially for those who already have chronic kidney diseases, such as diabetic nephropathy and nephrotic syndrome. Therefore, regular clinical follow-ups with healthcare professionals are essential for individuals consuming protein products. On the one hand, it is important to discuss the dosages of hemp protein intake with healthcare professionals, who can identify the best dietary intervention course for consumers. On the other hand, it is necessary to take into account the anti-nutritional compounds.

## 8. Conclusions and Future

Hemp seed is a novel protein-rich functional food with promising phytotherapy efficacy. The research and applications on hemp are relatively few compared with other plant foods. Thus, hemp intrigues the growing interest of scientists. This review focuses on the antioxidant ability of hemp seed protein and summarizes the characteristics of antioxidant peptides. It found that the structure-activity relationship of antioxidant peptides is impacted by multiple factors involving the molecular size, the compositions of amino acids and the sequences of amino acids. Considering the positive pre-clinical properties of hemp protein, many scholars offered a broad spectrum of potential pharmacology on various diseases such as hypertension, inflammatory diseases, cancer. Since hemp seed is still an underexploited food, more high-quality experiments such as proteomic analysis, genomics analysis, and the clinical trials of large sample size are warranted for more scientific evidence. Despite the current limitations of hemp protein products, we believe they will be better applied to human health worldwide with the development of food, agriculture, biology, and medicine technologies. We are looking forward to seeing a brilliant future of hemp protein.

## Figures and Tables

**Figure 1 molecules-27-07924-f001:**
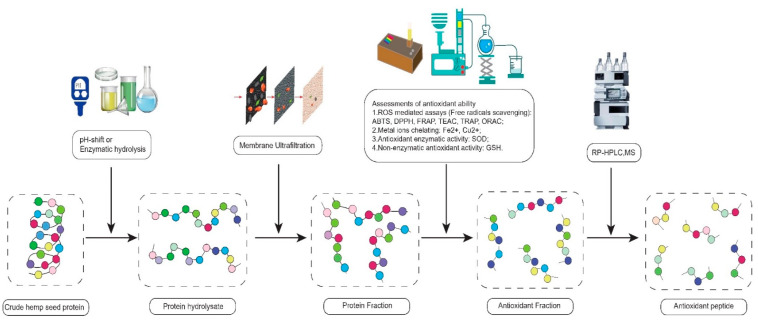
The experimental flow chart of isolation of antioxidant peptides.

**Figure 2 molecules-27-07924-f002:**
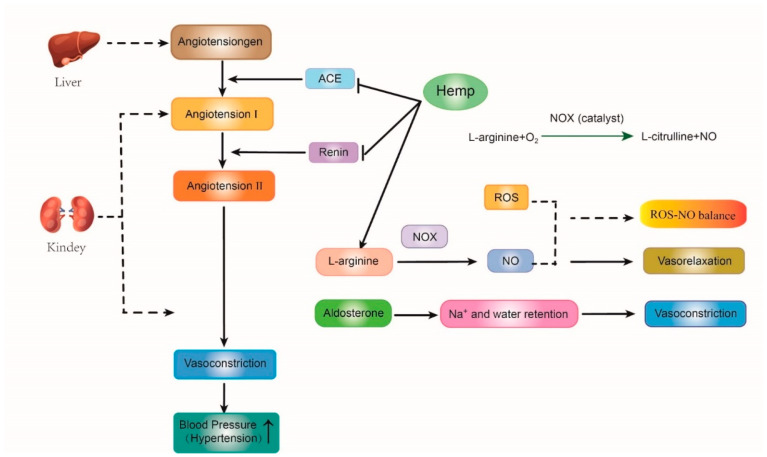
The mechanism of antioxidant and anti-hypertension properties of hemp bioactive peptides on the rennin-angiotensin-aldosterone system. ACE, angiotensin-converting enzyme; NOS, nitric oxide synthase; NO, nitric oxide; ROS, reactive oxygen species; 
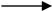
 represents promote or inactive; 
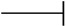
 represents inhibit or inactive.

**Figure 3 molecules-27-07924-f003:**
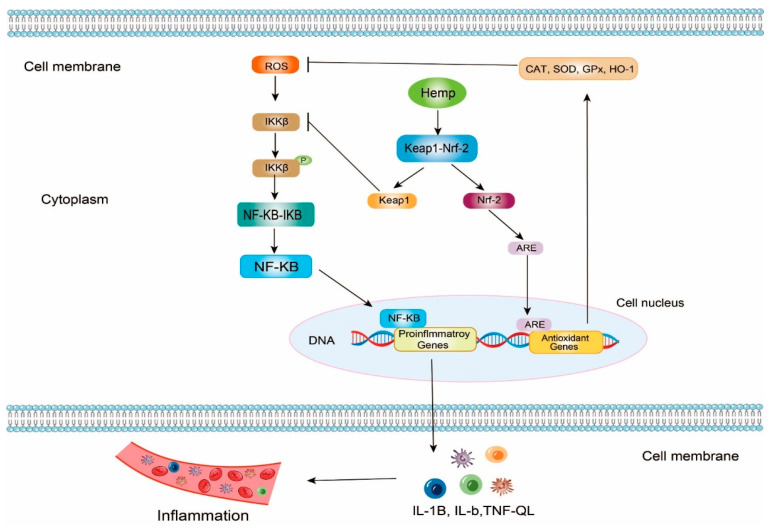
The mechanism of antioxidant and anti-inflammatory properties of hemp bioactive peptides on NF-κB and Keap1-Nrf2 signaling pathway. ROS, reactive oxygen species; IKKβ, inhibitor of κB Kinase β; NF-κB, Nuclear Factor κB; Keap1, Kelch-like ECH-associated protein 1; Nrf2, nuclear factor erythroid 2-related factor 2; ARE, antioxidant response elements; CAT, catalase; SOD, superoxide dismutase; GPx, glutathione peroxidase; HO-1, Heme Oxygenase-1; P, TNF, tumor necrosis factor; IL, interleukin, phosphorylation; 
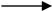
 represents promote or active; 
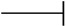
 represents inhibit or inactive.

**Figure 4 molecules-27-07924-f004:**
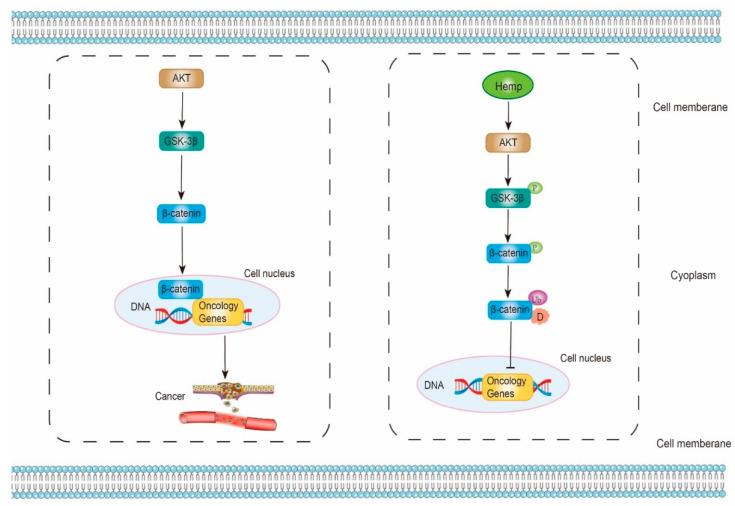
The mechanism of antioxidant and anti-cancer properties of hemp bioactive peptides on Akt/GSK-3β/β-catenin signaling pathway. Akt, protein kinase B; GSK-3β, glycogen synthase kinase-3β; P, phosphorylation; Ub, ubiquitination; D, degradation; 
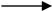
 represents promote or active; 
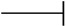
 represents inhibit or inactive.

**Figure 5 molecules-27-07924-f005:**
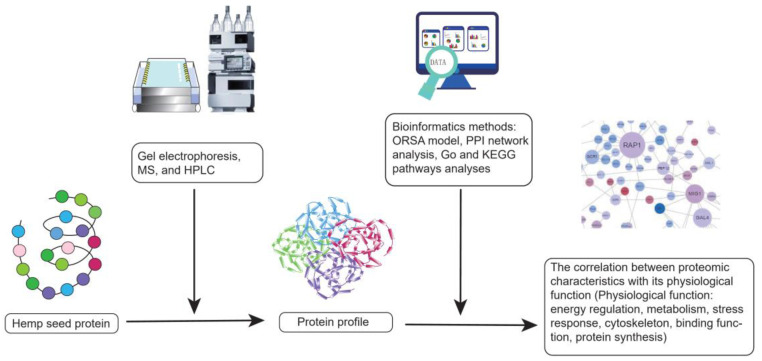
The experimental flow chart of protein profile.

**Table 1 molecules-27-07924-t001:** In vitro antioxidant assays of soybean and hemp protein hydrolysates.

	Soybean Hydrolysate	Hemp Hydrolysate	*p* Value
ORAC (μM trolox/g sample)	38.78 ± 0.362	0.539 ± 0.124	<0.001
Metal iron chelating activity (%)	42.71 ± 0.247	0.591 ± 0.152	<0.05

Note: Values are means ± standard deviations, *n* = 3.

**Table 2 molecules-27-07924-t002:** Amino acid compositions of hemp proteins and hydrolysates.

Amino	Defatted	Hemp Protein	Hemp Protein Isolate					Hemp Protein Fraction		
His	3.07 ± 0.01	2.78 ± 0.03	1.09 ± 0.33	2.23 ± 0.20	2.85 ± 0.05	2.81 ± 0.47	2.61 ± 0.06	2.47 ± 0.01	2.47 ± 0.03	2.49 ± 0.08
Arg	12.68 ± 0.21	14.07 ± 0.40	6.05 ± 0.56	10.64 ± 0.92	13.44 ± 0.03	9.91 ± 0.91	13.87 ± 0.11	12.96 ± 0.13	13.6 ± 0.25	13.31 ± 0.24
Thr	3.79 ± 0.08	3.68 ± 0.29	2.06 ± 0.18	3.36 ± 0.37	3.79 ± 0.10	4.57 ± 0.35	3.60 ± 0.04	3.77 ± 0.35	4.01 ± 0.19	4.00 ± 0.19
Val	5.27 ± 0.11	4.66 ± 0.02	2.51 ± 0.09	3.90 ± 0.12	3.77 ± 0.40	4.98 ± 0.13	5.67 ± 0.14	5.26 ± 0.11	4.45 ± 0.19	4.24 ± 0.14
Met	2.14 ± 0.11	1.81 ± 0.39	1.83 ± 0.15	1.63 ± 0.05	3.79 ± 0.10	1.39 ± 0.06	1.94 ± 0.04	2.03 ± 0.06	1.85 ± 0.14	1.80 ± 0.15
Lys	3.78 ± 0.09	2.97 ± 0.06	1.20 ± 0.23	1.97 ± 0.18	0.89 ± 0.34	4.16 ± 0.87	3.19 ± 0.11	2.94 ± 0.14	2.56 ± 0.03	2.51 ± 0.07
Ile	4.23 ± 0.08	3.84 ± 0.07	1.93 ± 0.02	3.12 ± 0.10	3.02 ± 0.40	3.99 ± 0.08	4.15 ± 0.13	4.16 ± 0.11	3.98 ± 0.05	3.90 ± 0.04
Leu	7.12 ± 0.05	6.75 ± 0.04	3.13 ± 0.08	4.90 ± 0.19	7.13 ± 0.11	6.63 ± 0.23	9.91 ± 0.06	7.26 ± 0.10	5.15 ± 0.04	4.82 ± 0.43
Phe	4.76 ± 0.09	4.60 ± 0.04	2.34 ± 0.19	3.86 ± 0.42	5.12 ± 0.02	4.57 ± 0.11	7.68 ± 0.05	5.01 ± 0.14	3.21 ± 0.02	2.85 ± 0.40
Non-essential										
Trp	0.76 ± 0.05	1.23 ± 0.16	0.04 ± 0.00	0.02 ± 0.00	0.55 ± 0.02	NA	1.58 ± 0.01	1.44 ± 0.15	1.16 ± 0.12	1.11 ± 0.02
Asp	11.34 ± 0.37	11.39 ± 0.03	4.06 ± 0.40	6.55 ± 0.48	12.35 ± 0.34	9.41 ± 0.39	9.49 ± 0.06	11.70 ± 0.29	12.79 ± 0.47	12.70 ± 0.11
Ser	5.57 ± 0.11	4.63 ± 0.09	2.37 ± 0.23	3.90 ± 0.31	6.31 ± 0.21	5.18 ± 0.02	4.73 ± 0.21	4.79 ± 0.19	4.69 ± 0.07	4.47 ± 0.38
Glu	18.41 ± 0.13	20.06 ± 1.34	6.92 ± 0.45	11.46 ± 0.82	18.86 ± 0.35	16.14 ± 0.26	15.18 ± 0.96	19.31 ± 1.04	22.71 ± 1.58	22.87 ± 1.02
Gly	4.74 ± 0.07	4.29 ± 0.23	2.01 ± 0.24	3.19 ± 0.29	4.79 ± 0.23	3.99 ± 0.06	3.23 ± 0.06	3.93 ± 0.07	4.54 ± 0.20	4.71 ± 0.44
Ala	4.89 ± 0.10	4.47 ± 0.16	1.79 ± 0.04	2.73 ± 0.09	5.01 ± 0.14	4.50 ± 0.36	4.91 ± 0.06	4.77 ± 0.12	4.30 ± 0.19	4.12 ± 0.09
Pro	2.66 ± 0.69	4.00 ± 0.07	1.73 ± 0.07	2.66 ± 0.10	3.02 ± 0.00	4.53 ± 0.39	3.19 ± 0.33	4.04 ± 0.56	4.23 ± 0.11	4.89 ± 0.73
Cys	1.22 ± 0.02	1.32 ± 0.23	0.71 ± 0.02	0.99 ± 0.06	0.89 ± 0.34	0.17 ± 0.01	0.29 ± 0.13	0.66 ± 0.01	1.26 ± 0.07	1.58 ± 0.28
Tyr	3.79 ± 0.08	3.45 ± 0.00	1.51 ± 0.13	2.71 ± 0.26	3.79 ± 0.10	3.67 ± 0.23	4.78 ± 0.02	3.50 ± 0.02	3.06 ± 0.06	3.62 ± 0.96
Reference	[76]	[59]	[77]	[77]	[76]	[7]	[59]	[59]	[59]	[59]

Note: Each value is the mean and standard deviation and all values are expressed in g of amino acid per 100 g of protein. NA is not available.

**Table 3 molecules-27-07924-t003:** Amino acid sequence of antioxidant peptides.

Protein Resource	Hydrolytic Enzyme	In Vitro Antioxidant Assays	Amino Acid Sequence	Reference
Sardine Muscle	Pepsin	Superoxide radical scavenging activity, hydroxyl radical scavenging activity.	Leu-Gln-Pro-Gly-Gln-Gly-Gln-Gln	[86]
Casein	Pepsin	Superoxide radical scavenging activity, hydroxyl radical scavenging activity, DPPH radical scavenging activity.	Tyr-Phe-Tyr-Pro-Glu-Leu	[87]
Royal Jelly	Protease N	Superoxide radical scavenging activity, hydroxyl radical scavenging activity, hydrogen peroxide scavenging activity, metal chelating activity.	Ala-Leu, PheLys, Phe-Arg, lle-Arg, Lys-Phe, Lys-Leu, Lys-Tyr, Arg-Tyr, Tyr-Asp, Tyr-Tyr, Leu-Asp-Arg, Lys-Asn-Tyr-Pro	[88]
Egg White	Pepsin	ORAC, low-density lipoprotein lipid oxidation induced by Cu^2+^.	Tyr-Ala-Glu-Glu-Arg-Tyr-Pro-Ile-Leu	[89]
Porcine Myofibrillar	Actinase E, Papain	DPPH radical scavenging activity, metal ion chelating activity, hydroperoxides in a peroxidation system.	Asp-Ser-Gly-Val-Thr, Ile-Glu-Ala-Glu-Gly-Glu, Asp-Ala-Gln-Glu-Lys-Leu-Glu, Glu-Glu-Leu-Asp-Asn-Ala-Leu-Asn, Val-Pro-Ser-Ile-Asp-Asp-Gln-Glu-Glu-Leu-Met	[90]
Sardinelle	Alcalase	DPPH radical-scavenging assay, the lipid peroxidation inhibition activity, reducing power assay.	Leu-Ala-Arg-Leu, Gly-Gly-Glu, Leu-His-Tyr and Gly-Ala-Leu-Ala-Ala-His	[78]
Barley	Alcalase, Flvourzyme and Pepsin	DPPH radical scavenging assay, superoxide radical scavenging assay, hydroxyl radical scavenging assay, metal chelating activity, ORAC.	Pro-Tyr-Pro, Gln-Pro-Tyr-Pro-Gln, Gln-Gln-Pro-Tyr-Pro-Gln, Gln-Pro-Gln-Pro-Tyr-Pro-Gln, Thr-Gln-Gln-Pro-Tyr-Pro-Gln, Glu-Pro-Tyr-Pro-Glu	[80]
Corn	Alcalase	Superoxide dismutase activity.	Leu-Asp-Tyr-Glu	[79]

**Table 4 molecules-27-07924-t004:** Amino acid compositions of dietary protein and the FAO/WHO/UNU suggested requirements.

Amino Acids	Hemp Seed	Brown Rice	Soybean	Brown Rice	Oat	Pea	Potato	Wheat	Fababean	Corn	Infant (FAO/WHO/UNU)	Child 2–5 Years Old (FAO/WHO/UNU)
Essential												
His	3.07	1.8	2.3	1.8	0.9	1.6	1.4	1.7	2.7	2.2	2.6	1.90
Arg	12.68	6.3	1.8	6.3	3.1	5.9	3.3	4.1	11.2	6.3	NA	NA
Thr	3.79	2.9	7.9	2.9	1.5	2.5	4.1	2.3	3.1	3.4	4.3	3.40
Val	5.27	4.6	5.9	4.6	2.0	2.7	3.7	3.6	6.8	7.3	5.5	3.50
Met	2.14	2.3	1.7	2.3	0.1	0.3	1.3	1.1	0.7	1.6	NA	NA
Lys	3.78	2.4	9.7	2.4	1.3	4.7	4.8	2.0	7.5	4.3	6.6	5.80
Ile	4.23	3.5	4.3	3.5	1.3	2.3	3.1	3.2	4.3	2.6	4.6	2.80
Leu	7.12	6.4	10.2	6.4	3.8	5.7	6.7	6.7	9.6	10.4	9.3	6.60
Phe	4.76	4.4	2.6	4.4	2.7	3.7	4.2	2.3	5.1	4.1	NA	NA
Non-essential												
Trp	0.76	1.2	1.1	1.2	NA	NA	NA	1.2	NA	NA	NA	NA
Asp	11.34	6.9	10.2	6.9	NA	NA	NA	4.5	7.6	7.1	NA	N
Ser	5.57	3.9	4.6	3.9	2.2	3.6	3.4	4.2	5.6	5.5	NA	NA
Glu	18.41	13.9	17.5	13.9	11.0	12.9	7.1	36.1	18.1	18.7	NA	NA
Gly	4.74	3.5	3.6	3.5	1.7	2.8	3.2	3.3	5.3	6.6	NA	NA
Ala	4.89	4.5	4.8	4.5	2.2	3.2	3.3	2.8	4.7	7.6	NA	NA
Pro	2.66	2.9	5.7	2.9	2.5	3.1	3.3	8.4	5.4	9.0	NA	NA
Cys	1.22	1.7	2.1	1.7	0.4	0.2	0.3	2.0	1.0	1.9	NA	NA
Tyr	3.79	4.3	1.1	4.3	1.5	2.6	3.8	4.0	1.3	1.4	1.7	1.10
Reference	[76]	[135]	[135]	[135]	[136]	[136]	[136]	[137]	[138]	[138]	[139]	[140]

Note: The values of soybean, rice and whey are presented in g per 100 g protein isolated. Other values are presented in g per 100 g raw material. NA is not available.

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
