# Peer review of "Antioxidant Properties of Hemp Proteins: From Functional Food to Phytotherapy and Beyond"

_molecules, 2022, doi:10.3390/molecules27227924_

Round 1

Reviewer 1 Report

Hemp attracts increasing attention from consumers and scientists as hemp products provide a wide spectrum of potential functions. Particularly, bioactive peptides derived from hemp proteins have been proven to be strong antioxidants, which is an extremely hot research topic in recent years. However, some controversial disputes and unknown issues are still underway to be explored and verified in the aspects of technique, methodology, characteristics, mechanisms, application, caution, etc. Therefore, this review focusing on the antioxidant properties of hemp proteins is necessary to discuss the multiple critical issues, including in vitro structure-modifying techniques and antioxidant assays, structure-activity relationships of antioxidant peptides, pre-clinical studies on hemp proteins and pathogenesis-related molecular mechanisms, usage and potential hazard, and novel advanced techniques involving bioinformatics methodology (QSAR, PPI, GO, KEGG), proteomic analysis, and genomics analysis, etc. The antioxidant potential of hemp proteins may provide both functional food benefits and phytotherapy efficacy to human health.

It is well organization and it provide interesting view to the author. However, the structure identification of the protein should be summarized, such as (Characterization and determination of bovine immunoglobulin G in milk and dairy products by ultra-high performance liquid chromatography mass spectrometry. Food Chemistry, 390:133170.)

There are also some other modification technologies such as glycosylation, while analyze by UPLC-MS/MS method (Quantitative determination of Nepsilon-(carboxymethyl)lysine in sterilized milk by isotope dilution UPLC-MS/MS method without derivatization and ion pair reagents. Food chemistry. 385(2022): 132697).

In “6. Hemp Seed Protein Product and Usage”

The hemp seed also possess phytochemicals such as polyphenols that fortification the protein function (The positive correlation of antioxidant activity and prebiotic effect about oat phenolic compounds. Food Chemistry, 402(2023): 134231).

Author Response

It is well organization and it provides interesting view to the author. However, the structure identification of the protein should be summarized, such as (Characterization and determination of bovine immunoglobulin G in milk and dairy products by ultra-high performance liquid chromatography mass spectrometry. Food Chemistry, 390:133170.)

Response: Per suggested, additional sentences regarding structure identification of the protein and a new reference #82 have been added in Lines 262-268.

There are also some other modification technologies such as glycosylation, while analyze by UPLC-MS/MS method (Quantitative determination of Nepsilon-(carboxymethyl)lysine in sterilized milk by isotope dilution UPLC-MS/MS method without derivatization and ion pair reagents. Food chemistry. 385(2022): 132697).

Response: Other modification technologies and a new reference #83 have also been summarized in Lines 262-268.

In “6. Hemp Seed Protein Product and Usage”

The hemp seed also possess phytochemicals such as polyphenols that fortification the protein function (The positive correlation of antioxidant activity and prebiotic effect about oat phenolic compounds. Food Chemistry, 402(2023): 134231).

Response: “6. Hemp Seed Protein Product and Usage” is focused on hemp seed protein rather than oat or phytochemical phenolics. However, we agree the hemp seed possesses phytochemical phenolics that can be included as an additional reference #81 in Lines 262-268.

Reviewer 2 Report

The article presented for review contains very important and necessary information about the antioxidant properties of hemp proteins. The article collects a lot of information on this subject, using modern scientific literature. The article is a review, there is very little of my own research and observation, and if any, only rudely discussed. My comments concern:

1. Table 1 presents a pilot experiment on the antioxidant capacity of hemp and soy protein - why is the methodology for determining these parameters not provided?

2. I have similar remarks to Table 2 and their results, why were they not discussed, even briefly and to the point?

Author Response

The article presented for review contains very important and necessary information about the antioxidant properties of hemp proteins. The article collects a lot of information on this subject, using modern scientific literature. The article is a review, there is very little of my own research and observation, and if any, only rudely discussed. My comments concern:

1. Table 1 presents a pilot experiment on the antioxidant capacity of hemp and soy protein - why is the methodology for determining these parameters not provided?

Response: Per suggested, the methodology for Table 1’s parameters has been added in Lines 161-162.

2. I have similar remarks to Table 2 and their results, why were they not discussed, even briefly and to the point?

Response: Per suggested, additional discussion about Table 2 has been added in lInes 230-237.

Reviewer 3 Report

Plese find below reviewers suggestion of the manuscript: Antioxidant properties of hemp proteins: from functional food to phytotherapy and beyond

Different font and size is used in same paragraph through manuscript

Introduction should be extended and  should contain the aim of the research

Line 56 pH both letters are cappital

Figure 1. The experimental flow chart of isolation of antioxidant peptides.

It is not clear are the results presented in table 2 results of the authors or are collected from literature.

The part where methods are described should be extended in the way of application for hemp.

Second part of manuscript is well written and literature is covered.                

Author Response

Plese find below reviewers suggestion of the manuscript: Antioxidant properties of hemp proteins: from functional food to phytotherapy and beyond.

Different font and size is used in same paragraph through manuscript

Response: The different font and size of Lines 41-48 and Lines 108-116, the format of Table 2’s first line, the format of the entire Table 3, the format of Figure 2’s annotation, and the format of Table 4’s note have been modified.  

Introduction should be extended and  should contain the aim of the research

Response: Per suggested, additional paragraph including the aim of the research has been added in Lines 49-58.

Line 56 pH both letters are cappital

Response: Per suggested, Line 56 (Line 65 in the revised version) pH has been corrected.

Figure 1. The experimental flow chart of isolation of antioxidant peptides.

Response: Per suggested, the Figure 1 title has been corrected.

It is not clear are the results presented in table 2 results of the authors or are collected from literature.

Response: The results of Table 2’s are summarized from the literature that have been denoted as reference in Table 2.

The part where methods are described should be extended in the way of application for hemp.

Response: Per suggested, additional paragraph regarding potential application for hemp has been added in Lines 117-124.

Second part of manuscript is well written and literature is covered.    

Round 2

Reviewer 1 Report

The antioxidant potential of hemp proteins may provide both functional food benefits and phytotherapy efficacy to human health. It is well organization and it provide interesting view to the author. In the  “Conclusions and Future” part, antioxidative and gut microbiota regulation is a good concerns (Whole grain benefit: oat β-glucan and phenolic compounds synergistically regulates hyperlipidemia via gut microbiota in high-fat-diet mice. Food & Function, 2022, Doi: 10.1039/D2FO01746F).

Reviewer 2 Report

The work after the corrections is acceptable for publication in this publishing house.